# The Potential of 2-aza-8-Oxohypoxanthine as a Cosmetic Ingredient

**Hisae Aoshima [1,*], Masayuki Ito [1], Rinta Ibuki [1] and Hirokazu Kawagishi [2,3,4,*]**

[1] Department of Research and Development, Vitamin C60 BioResearch Corp, Nihonbashidori-Nichoume Bldg, 4F 2-2-6 Nihonbashi Chuo-ku, Tokyo 103-0027, Japan; masayuki.ito@vc60.com (M.I.); rinta.ibuki@vc60.com (R.I.)

[2] Research Institute of Green Science and Technology, Shizuoka University, 836 Ohya, Suruga-ku, Shizuoka 422-8529, Japan

[3] Graduate School of Science and Technology, Shizuoka University, 836 Ohya, Suruga-ku, Shizuoka 422-8529, Japan

[4] Graduate School of Integrated Science and Technology, Shizuoka University, 836 Ohya, Suruga-ku, Shizuoka 422-8529, Japan

[*] Correspondence: hisae.aoshima@vc60.com (H.A.); kawagishi.hirokazu@shizuoka.ac.jp (H.K.)

**Abstract:** In this study, we verified the effects of 2-aza-8-oxohypoxanthine (AOH) on human epidermal cell proliferation by performing DNA microarray analysis. Cell proliferation was assessed using the 3-(4,5-dimethylthiazol-2-yl)-2,5-diphenyltetrazolium bromide assay, which measures mitochondrial respiration in normal human epidermal keratinocyte (NHEK) cells. Gene expression levels were determined by DNA microarray analysis of 177 genes involved in skin aging and disease. AOH showed a significant increase in cell viability at concentrations between 7.8 and 31.3 μg/mL and a significant decrease at concentrations above 250 μg/mL. DNA microarray analysis showed that AOH significantly increased the gene expression of CLDN1, DSC1, DSG1, and CDH1 (E-cadherin), which are involved in intercellular adhesion and skin barrier functioning. AOH also up-regulated the expression of KLK5, KLK7, and SPIMK5, which are proteases involved in stratum corneum detachment. Furthermore, AOH significantly stimulated the expression of KRT1, KRT10, TGM1, and IVL, which are considered general differentiation indicators, and that of SPRR1B, a cornified envelope component protein. AOH exerted a cell activation effect on human epidermal cells. Since AOH did not cause cytotoxicity, it was considered that the compound had no adverse effects on the skin. In addition, it was found that AOH stimulated the expression levels of genes involved in skin barrier functioning by DNA microarray analysis. Therefore, AOH has the potential for practical use as a cosmetic ingredient. This is the first report of efficacy evaluation tests performed for AOH.

**Keywords:** fairy chemical; 2-aza-8-oxohypoxanthine; microarray analysis; skin barrier function

## 1. Introduction

Turfgrass occasionally grows in the shape of a ring, with thicker growth than the surrounding grass, and thereafter, mushrooms form in the same circle (Figure 1). This phenomenon, which occurs on lawns in parks and golf courses, is called "fairy rings." In Western legend, it is said that fairies make these circles and dance inside of them [1–3]. Since the first paper on "fairy rings" published in 1675 and the subsequent papers were introduced in Nature in 1875, the cause of the phenomenon had remained a mystery for a long time until our finding on "fairy chemicals" [4]. We cultured the hyphae of one of the fairy-ring-forming fungi, *Lepista sordida*, and discovered a plant growth stimulator, 2-azahypoxanthine (AHX), in the culture broth. We also found imidazole-4-carboxamide (ICA), which suppressed the growth of turfgrass from the broth [5,6]. Furthermore, it has been revealed that AHX was converted into 2-aza-8-oxohypoxanthine (AOH) when absorbed into plants (Figure 2) [7]. We named AHX, ICA, and AOH as fairy chemicals

(FCs) after the title of the article in *Nature* that covered our study [8]. Subsequent studies have demonstrated that FCs endogenously exist in all the plants tested, including edible parts of three major cereal crops, rice, wheat, and corn [7]; thus, people have eaten FCs for a long time. Furthermore, FCs increased the yields of rice, wheat, and other crops under field and greenhouse conditions [5,9,10]. These results suggest that FCs are a new family of plant hormones [11,12].

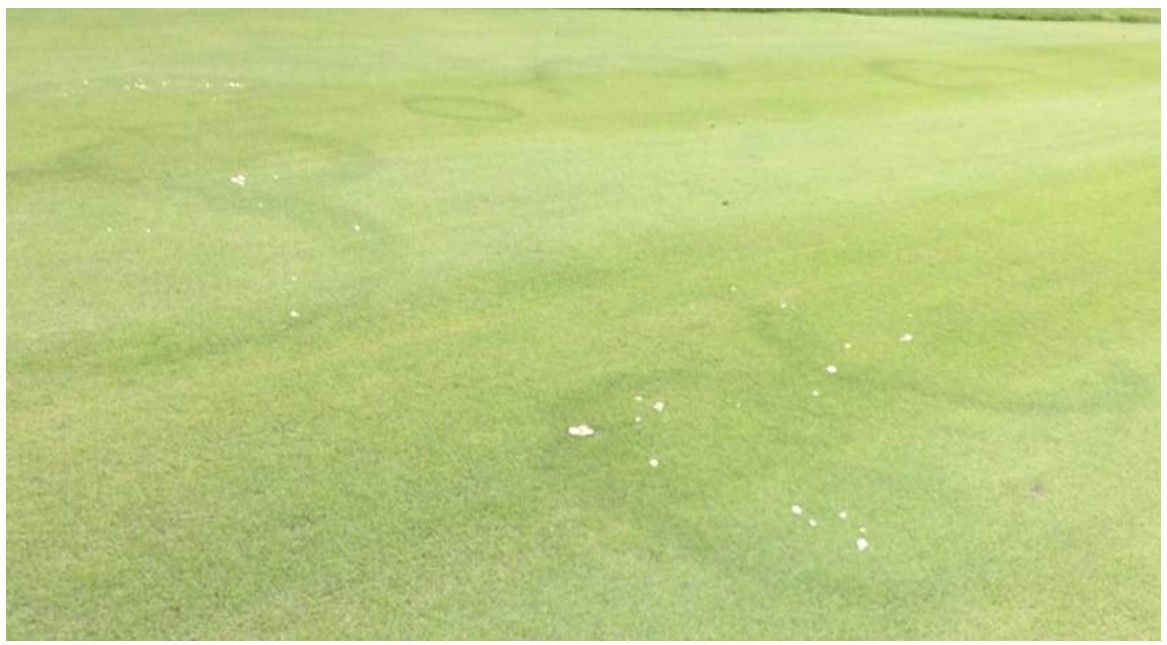

**Figure 1.** Fairy rings appearing on golf course grass.

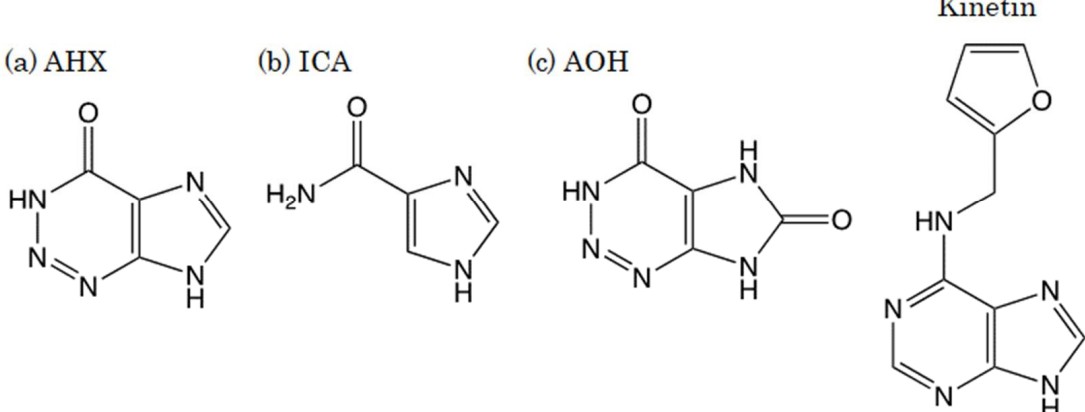

**Figure 2.** Structures of (**a**) 2-azahypoxanthine (AHX), (**b**) imidazole-4-carboxamide (ICA), (**c**) 2-aza-8-oxohypoxanthine (AOH), and kinetin.

Kinetin (N6-furfuryladenine) is a member of the cytokinin plant growth hormone family and is known for its growth-promoting and anti-aging effects on plants (Figure 2). Stanley B. Levy of the Revlon Consumer Products Corporation stated that kinetin is also useful against human skin aging [13]. Many studies have reported the anti-aging effects of kinetin on human skin cells in clinical trials [14–18]. Therefore, kinetin is currently used as an ingredient in cosmetics and cosmeceuticals. Similar to kinetin, FCs also have a purine carbon skeleton (Figure 2) and regulate plant growth. Therefore, we thought that FCs might have similar effects to kinetin on human skin cells and examined them. First of all, the safety of FCs was evaluated. AOH yielded the best result and showed no toxicity in

any of the safety evaluations for cosmetic applications, including Ames test, skin irritation, skin sensitization (DPRA), in vitro phototoxicity tests, and human patch tests [19]. In this study, we focused on AOH and evaluated the viability of normal human epidermal keratinocyte (NHEK) cells to verify the potential of AOH as a cosmetic ingredient. In addition, DNA microarray analysis was performed to analyze the effects of AOH on the skin comprehensively.

## 2. Materials and Methods

### 2.1. Chemicals and Reagents

AOH was synthesized from 5-aminoimidazole-4-carboxamide as previously described [7,20,21]. NHEKs (Kurabo Industries Ltd., Osaka, Japan) were cultured in HuMedia KG2 (KG2) and HuMedia KB2 (KB2) media (Kurabo Industries Ltd.). 3-(4,5-dimethylthiazol-2-yl)-2,5-diphenyltetrazolium bromide (MTT) and 2-propanol were purchased from Kanto Chemical Co., Inc. (Tokyo, Japan). QIAzol Lysis Reagent and an miRNeasy Mini Kit were purchased from QIAGEN (Hilden, Germany). MessageAmpII (biotin amplification kit) and DNA fragmentation reagent from Applied Biosystems Japan (Tokyo, Japan) were used. Streptavidin-cyanine 5 (Cy5) was purchased from GE Healthcare Bio-Science KK (Tokyo, Japan). For the mRNA expression analysis chip, a Genopal® NDR custom DNA chip from Mitsubishi Chemical Corporation (Tokyo, Japan) was used.

### 2.2. Cell Viability Assay

NHEK cells were seeded at a density of $5.0 \times 10^3$ cells in 96-well plates and incubated in KG2 medium at 37 °C and 5% $CO_2$ for 24 h. The cells were then incubated in KB2 medium containing AOH (0 to 500 μg/mL) for 48 h. KG2 medium was used as a positive control. NHEK cell viability was evaluated using the MTT assay. The cultured cells were replaced with KB2 medium containing 0.2 mg/mL MTT and incubated at 37 °C for 1 h. After removing the medium, 2-propanol (150 μL) was added to the cultured and lysed cells. Cell viability was evaluated by measuring the absorbance at 550 and 650 nm using an EnSpire microplate reader (PerkinElmer, Inc., Waltham, MA, USA). The amount of MTT reduction was determined by subtracting the absorbance at 650 nm from the absorbance at 550 nm. The amount of MTT reduction is indicated as a percentage (%) relative to the absorbance of the positive control. The Student's *t*-test was used to estimate differences between means at a 5% level of significance.

### 2.3. DNA Microarray Analysis

#### 2.3.1. Cell Culture and Treatment with AOX

NHEK cells were seeded at a density of $1.5 \times 10^5$ cells in 6-well plates and incubated in the KG2 medium for 24 h. The cells were then incubated in KB2 medium (3 mL) containing AOH (0, 30, 100, and 300 μg/mL) for 24 h ($n = 3$). After incubation, NHEK cells were lysed using QIAzol Lysis Reagent. Total RNA was extracted and purified from the NHEK lysis solution using the miRNeasy Mini Kit according to the manufacturer's instructions.

#### 2.3.2. RNA Isolation

The concentration of the obtained total RNA was measured using a Nanodrop spectrophotometer (Thermo Fisher Scientific, Wilmington, DE, USA).

The quality of the RNA samples used for the microarray analysis was examined using the RNA 6000 Nano LabChip Kit (p/n 5067-1511) on an Agilent 2100 Bioanalyzer (G2938C; Agilent Technologies, Inc., Palo Alto, CA, USA). RNA was stored in RNase-free water at −80 °C. RNA with an RNA integrity number of nine or more was used for further experiments.

#### 2.3.3. Microarray Analysis

First, RNA was amplified using the MessageAmpII biotin-enhanced amplification kit according to the manufacturer's instructions and column purified. Biotinylated mRNA

(5 μg) was fragmented using fragmentation reagent and then incubated at 94 °C for 7.5 min [22]. The fragmentation reaction was terminated by adding a stop solution. Hybridization was carried out with a DNA microarray Genopal® in 180 mL of hybridization buffer (0.12 mol $L^{-1}$ Tris-HCl/0.12 mol $L^{-1}$ NaCl and 0.05% Tween-20) and 5 mg of fragmented biotinylated mRNA at 65 °C overnight. After hybridization, the DNA microarray was washed twice in wash solution A (0.12 mol $L^{-1}$ Tris-HCl/0.12 mol $L^{-1}$ NaCl and 0.05% Tween-20), followed by washing in wash solution B (0.12 mol $L^{-1}$ Tris-HCl/0.12 mol $L^{-1}$ NaCl). The DNA microarray was labeled with streptavidin-Cy5. The fluorescent-labeled DNA microarray was washed 4 times in wash solution B at room temperature for 5 min. Hybridization signal acquisition was performed using a DNA microarray reader with multibeam excitation technology (Yokogawa Electric Co., Tokyo, Japan) [23]. The DNA microarrays were scanned at multiple exposure times ranging from 0.1 to 40 s, and the intensity values with the best exposure conditions for each spot were selected. The obtained results were analyzed, and the results of various gene expression levels were represented in which the untreated control was 1. Statistical analysis was performed on the expression ratios using a paired Student's *t*-test.

### 2.4. Statistical Analysis

Data are expressed as the arithmetic mean ± SEM. Significant differences between the three groups were determined by Welch's one-way ANOVA followed by the Tukey's post hoc test using the statistical software SAS 9.4 (SAS Institute Japan Ltd., Tokyo, Japan) (GLM procedure with Welch option), the freeware R 3.0.2 (http://www.r-project.org/, accessed on 18 December 2018, Vienna, Austria, qtukey function), and Excel 2013 (Microsoft, Redmond, WA, USA). Two-group comparisons were analyzed with Student's *t*-test using GraphPad Prism 5 for Windows Ver. 5.01 (GraphPad Software, San Diego, CA, USA). *p*-values < 0.05 were considered significant.

## 3. Results

### 3.1. Cell Viability Assay

The MTT assay was used to evaluate the effect of AOH on the proliferation of NHEK cells. NHEK cells were treated with AOH (0 to 500 μg/mL) and incubated at 37 °C for 48 h. As shown in Figure 3, AOH showed a significant increase in cell viability at concentrations of 7.8–31.3 μg/mL.

### 3.2. DNA Microarray Analysis

The MTT assay showed that AOH had a cell-activating effect at a concentration of 31.3 μg/mL or lower for 48 h incubation (Figure 3). On the other hand, when 250 μg/mL or more of AOH was added, cytotoxicity was observed. However, when the incubation time was 24 h, no cytotoxicity was observed at 500 μg/mL or less, so for the following experiments, the incubation time after the addition of AOH was 24 h. To comprehensively analyze the effects of AOH on skin cells, DNA microarray analysis was performed on 177 skin-related genes. Among these genes, the expression of those involved in barrier functioning of the skin was increased by treatment with AOH (Table 1). The expression of claudin-1 (CLDN1), desmocollin-1 (DSC1), and desmoglein-1 (DSG1), which play important roles in the construction of tight junctions in the stratum corneum, was up-regulated 1.43-, 1.59-, and 1.15-fold, respectively, when AOH (300 μg/mL) was added to the culture. The expression of E-cadherin (CDH1), which is involved in cell adhesion, increased by 1.16-fold. The expression of kallikrein-related peptidase (KLK)5 and KLK7, which encode serine proteases that promote stratum corneum exfoliation, increased 1.24- and 1.17-fold, respectively (300 μg/mL). Serine protease inhibitor Kazal-type (SPINK)5 and SPINK9 expressions were also increased 1.38- and 1.13-fold at 300 and 100 μg/mL concentrations of AOH, respectively. The expression of keratin (KRT)1 and KRT10, which are differentiation indicators of keratinocytes, increased 2.07- and 1.44-fold, respectively. BLMN, which encodes a protein that promotes the production of natural moisturizing factor, was

up-regulated 1.11-fold by treatment with 100 μg/mL AOH. AOH also up-regulated trans-glutaminase 1 (TGM1), involucrin (IVL), and small proline-rich protein 1B (SPRR1B) by 1.57-, 1.75-, and 1.6-fold, respectively. These proteins are involved in the formation of the cornified envelope (CE). The expression of hyaluronan synthase (HAS)3, which is involved in hyaluronan synthesis in the epidermis, was increased 1.6-fold by AOH at 300 μg/mL.

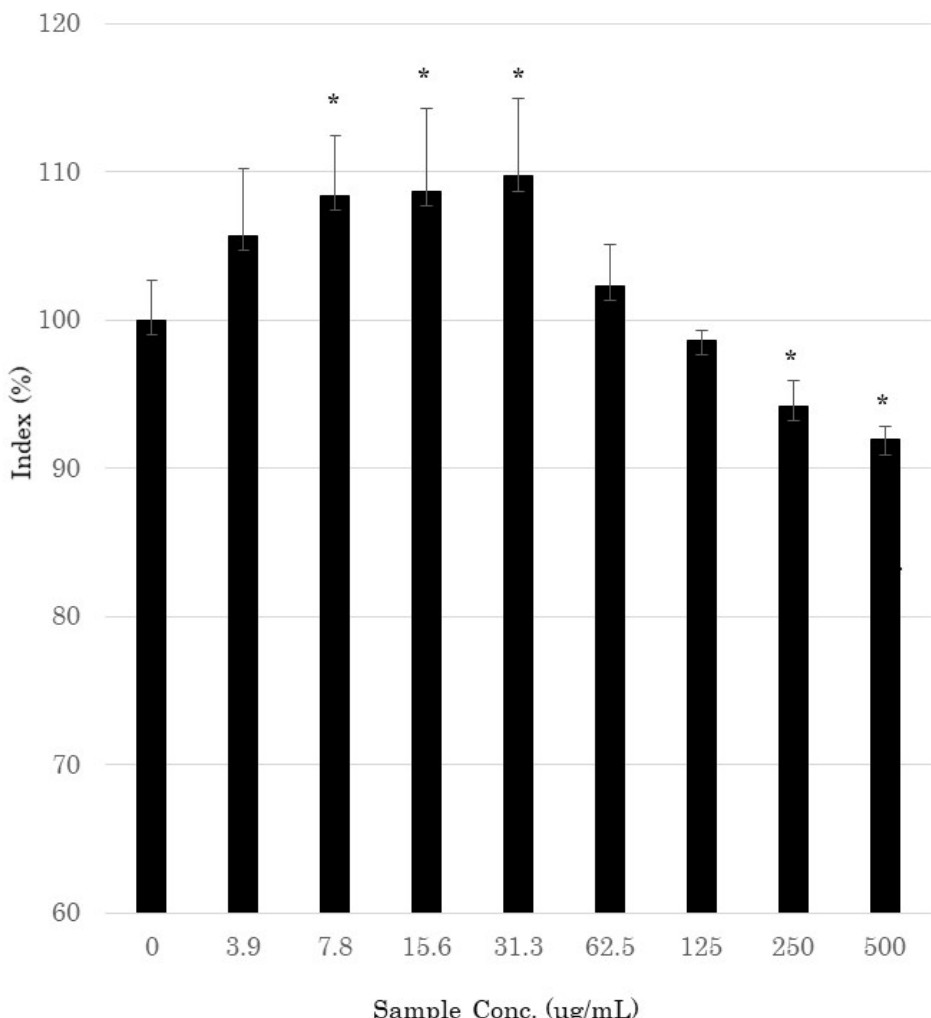

**Figure 3.** Cytotoxic effects of 2-aza-8-oxohypoxanthine (AOH) on normal human epidermal keratinocytes, as determined by the 3-(4,5-dimethylthiazol-2-yl)-2,5-diphenyltetrazolium bromide assay. The percent cell viability was calculated relative to the untreated control. * Significantly different from the control ($p < 0.05$; Student's $t$-test).

**Table 1.** Up-regulated genes involved in skin barrier functioning in normal human epidermal keratinocyte cells treated with 2-aza-8-oxohypoxanthine (AOH).

| Function | Effect | Gene Symbol | Gene Name (*Homo sapiens*) | Cont. | | AOH (30 µg/mL) | | AOH (100 µg/mL) | | AOH (300 µg/mL) | |
|---|---|---|---|---|---|---|---|---|---|---|---|
| | | | | Mean ± S.D. | *p* | Mean ± S.D. | *p* | Mean ± S.D. | *p* | Mean ± S.D. | *p* |
| Intercellular adhesion | Formation of tight junctions | CLDN1 | *Claudin-1* | 1.00 ± 0.12 | 1.000 | 1.19 ± 0.08 | 0.090 | 1.49 ± 0.07 | 0.006 | 1.43 ± 0.18 | 0.031 |
| | | DSC1 | *Desmocollin-1* | 1.00 ± 0.06 | 1.000 | 1.26 ± 0.23 | 0.170 | 1.28 ± 0.01 | 0.012 | 1.59 ± 0.09 | 0.001 |
| | | DSG1 | *Desmocollin-1* | 1.00 ± 0.02 | 1.000 | 1.09 ± 0.14 | 0.363 | 1.13 ± 0.04 | 0.020 | 1.15 ± 0.01 | 0.000 |
| | Cell adhesion factor (E-cadherin) | CDH1 | *Desmoglein-1* | 1.00 ± 0.07 | 1.000 | 1.11 ± 0.08 | 0.152 | 1.09 ± 0.04 | 0.130 | 1.16 ± 0.03 | 0.038 |
| Stratum corneum differentiation | Stratum corneum protease | KLK5 | *Homo sapiens Kallikrein-5* | 1.00 ± 0.02 | 1.000 | 1.11 ± 0.07 | 0.087 | 1.16 ± 0.02 | 0.001 | 1.24 ± 0.06 | 0.013 |
| | | KLK7 | *Homo sapiens Kallikrein-7* | 1.00 ± 0.08 | 1.000 | 1.11 ± 0.21 | 0.430 | 1.24 ± 0.05 | 0.016 | 1.17 ± 0.07 | 0.045 |
| | Inhibitor of stratum corneum protease | SPINK5 (LEKTI1) | *Serine peptidase inhibitor Kazal type 5* | 1.00 ± 0.04 | 1.000 | 1.15 ± 0.14 | 0.197 | 1.28 ± 0.03 | 0.002 | 1.38 ± 0.07 | 0.030 |
| | | SPINK9 (LEKTI2) | *Serine peptidase inhibitor Kazal type 9* | 1.00 ± 0.03 | 1.000 | 1.02 ± 0.16 | 0.859 | 1.13 ± 0.02 | 0.004 | 1.09 ± 0.08 | 0.176 |
| Epidermal differentiation | Filaggrin biosynthesis | BLMH | *Bleomycin hydrolase* | 1.00 ± 0.02 | 1.000 | 0.98 ± 0.06 | 0.648 | 1.11 ± 0.03 | 0.009 | 1.10 ± 0.05 | 0.068 |
| | Induction of stratum corneum differentiation | KRT1 | *Keratin 1* | 1.00 ± 0.04 | 1.000 | 1.468 ± 0.16 | 0.033 | 1.60 ± 0.11 | 0.006 | 2.07 ± 0.25 | 0.016 |
| | Induction of stratum corneum differentiation | KRT10 | *Keratin 10* | 1.00 ± 0.03 | 1.000 | 1.19 ± 0.07 | 0.034 | 1.33 ± 0.08 | 0.010 | 1.44 ± 0.14 | 0.029 |
| | Formation of CE | TGM1 | *Transglutaminase 1* | 1.00 ± 0.10 | 1.000 | 1.16 ± 0.14 | 0.173 | 1.29 ± 0.05 | 0.019 | 1.57 ± 0.08 | 0.002 |
| | | IVL | *Involucrin* | 1.00 ± 0.04 | 1.000 | 1.35 ± 0.16 | 0.055 | 1.49 ± 0.03 | 0.000 | 1.75 ± 0.11 | 0.004 |
| | | SPRR1B | *Small proline-rich protein 1 B* | 1.00 ± 0.04 | 1.000 | 1.27 ± 0.05 | 0.003 | 1.42 ± 0.05 | 0.000 | 1.60 ± 0.14 | 0.012 |
| Biosynthesis of hyaluronic acid | Hyaluronic acid synthase (epidermis) | HAS3 | *Hyaluronan synthase 3* | 1.00 ± 0.11 | 1.000 | 1.09 ± 0.19 | 0.510 | 1.27 ± 0.08 | 0.030 | 1.60 ± 0.13 | 0.004 |

Relative mRNA levels were calculated and compared to the control. Data are expressed as means ± standard deviations (SDs; $n = 3$). Values with significant differences from the control ($p < 0.05$; Student's *t*-test) are shown in red. CE: Cornified envelope.

## 4. Discussion

To investigate the effects of AOH on NHEK cells, cell viability was measured, and the results showed that AOH significantly increased cell viability at concentrations ranging from 7.8 to 31.3 µg/mL. This suggests that AOH might have an activating effect on human skin.

To comprehensively examine the possible effects of AOH on human skin, DNA microarrays were performed using NHEK cells. The addition of AOH stimulated the expression of CLD1, DSC1, DSG1, and CDH1, which are important for tight junction formation and intercellular adhesion. This result suggests that AOH enhances cell-cell interactions and strengthens adhesion, thereby improving the barrier function of the skin. Given that the expressions of both the protease gene group (KLK5 and KLK9), which is involved in stratum corneum exfoliation, and the protease inhibitor gene group (SPINK5, SPINK9) were up-regulated, AOH might promote skin turnover. AOH significantly increased the expression of KRT1 and KRT10, which may contribute to the strengthening of the epidermal cytoskeleton. It is known that, along with KRT10, which encodes a fibrin protein and a fibrillogenesis pair, KRT1 is involved in cytoskeleton formation in the upper epidermal cells of the skin and plays an important role in maintaining the mechanical strength of these cells. It has been reported that loss of KRT1 not only causes epidermal detachment but also affects the expression of pro-inflammatory cytokines, such as interleukin-18 [24]. AOH also up-regulated IVL and KRT1 expression levels. IVL and TGM1 are important substrates and enzymes for CE composition, which contributes to the physical or chemical toughness of the stratum corneum. An et al. reported that kinetin increases IVL and KRT1 expression in HaCaT cells using quantitative real-time polymerase chain reaction [25]. Kinetin, a plant hormone that promotes cell division, is used in cosmetics and medicinal cosmetics.

SPRR1B is also a necessary factor in the formation of CE membranes. However, a comparison of SPRR1B expression in non-lesional and lesional areas of atopic patients showed that SPRR1B expression in lesional areas was significantly higher than that in non-lesional areas [26]. However, since there are few studies on SPRR1B, it is difficult to consider the extent to which the increase in SPRR1B expression by AOH has an effect on the skin. Results of a double-blind clinical trial showed that eighr weeks of AOH application resulted in increased stratum corneum water content and decreased transepidermal water loss (TEWL) in the AOH group compared to the placebo group or before the start of the study (paper in preparation for submission). The clinical trial results seem to support the DNA microarray results.

Three major genes are known to synthesize hyaluronan, which is important for skin hydration: HAS1 and HAS2, which are localized in fibroblasts, and HAS3, which is responsible for hyaluronan synthesis in epidermal cells. AOH stimulated HAS3 expression but did not affect HAS1 and HAS2 expressions. This suggests that AOH may promote the biosynthesis of HA in the epidermis. These results suggest that AOH has a broad range of effects on epidermal functions, such as the promotion of epidermal turnover, differentiation, maturation, and the metabolism of dead skin cells.

## 5. Conclusions

This is the first study to investigate the efficacy of AOH for external skin use. We found that AOH significantly increased the viability of NHEK cells. The microarray results showed that AOH increased the expression of genes involved in skin barrier functions, such as intercellular adhesion, stratum corneum exfoliation, and differentiation induction. Furthermore, the expression of HAS3, a type of hyaluronan synthase, was also increased by treatment with AOH. Our study suggests that AOH may improve skin barrier functioning. Based on this research and previous studies, we conclude that AOH will be useful for external application on the skin and as a cosmetic ingredient to improve skin barrier functions [19]. In the future, it will be necessary to elucidate the mechanism with which AOH improves skin barrier functions.

**Author Contributions:** Conceptualization, H.A. and R.I.; methodology, H.A.; investigation, H.A. and M.I.; resources, H.A. and H.K.; data curation, H.A. and R.I.; writing—original draft preparation, H.A.; writing—review and editing, R.I. and H.K.; visualization, H.A.; supervision, H.K.; project administration, H.A.; funding acquisition, H.K. All authors have read and agreed to the published version of the manuscript.

**Funding:** This study was funded by Vitamin C60 BioResearch Corporation. This work was partially supported by a Grant-in-Aid for Scientific Research on Innovative Areas ("Frontier Research on Chemical Communications"; JP17H06402) from MEXT and a Grant-in-Aid for Specially Promoted Research "Science of fairy chemicals and their application development" (JP20H05620) from JSPS.

**Institutional Review Board Statement:** Not applicable.

**Informed Consent Statement:** Not applicable.

**Data Availability Statement:** Not applicable.

**Acknowledgments:** We thank K. Shimizu of COSMOS TECHNICAL CENTER (Tokyo, Japan) and M. Hayashi of Nikoderm Research Inc. (Osaka, Japan) for performing the cell viability assay and DNA microarray analysis.

**Conflicts of Interest:** This study was funded by Vitamin C60 BioResearch Corporation. Hisae Aoshima, Masayuki Ito, and Rinta Ibuki are employees of Vitamin C60 BioResearch Corporation.

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
