# Peer review of "The Potential of 2-aza-8-Oxohypoxanthine as a Cosmetic Ingredient"

_cosmetics, doi:10.3390/cosmetics8030060_

Round 1
Reviewer 1 Report
The paper entitled "The potential of 2-aza-8-oxohypoxanthine as a cosmetic ingredient" examines an somewhat interesting chemical isolated from plants that come into contact with mushroom fungi. The initial introduction was well written and while the term "Fairy Chemical" is somewhat colloquial, it does describe the somewhat unique relationship to the newly discovered chemical. the authors do appear to have a molecule similar to kinetin that may have some potential for use in topicals.
However, the authors have a significant problem in this paper as they appear to keep switching their units of measure from μg/ml to mg/ml. Importantly, the authors suggest that, based on the results of their MTT cell viability assays, the AOH becomes cytotoxic above 31.3 μg/ml (Figure 3). But, in their array data summary (Table 1) they suggest they ran the arrays at 100 and 300 μg/ml which would be 3-100 fold excess of the cytotoxic levels noted. And, throughout the paper the authors appear to keep switching between μg/ml and mg/ml. These units are not interchangeable! The authors either have made significant typographical errors or, they are not reporting their data correctly.
Regarding the array work, it is unclear from the Method section how the authors determine their statistical significance. It would be inappropriate to run the same set of cells three times and report a standard deviation in the measurement. To determine array standard deviations, the authors would need to run three or more separate arrays, using the same cells, treatments and data points, and then average and determine significance. This does not appear to be what they are doing (or it is unclear that this is what they are doing). Outside of this kind of analysis on arrays, it can only be determine subjectively that the treatments are having an impact on the cell genomic expressions.
This paper has some interesting ideas, but the significant problems above must be addressed before it can be accepted for publication.
Author Response
Thank you very much for reviewing our article. We apologize for the delay in response.
Point 1: However, the authors have a significant problem in this paper as they appear to keep switching their units of measure from μg/ml to mg/ml. Importantly, the authors suggest that, based on the results of their MTT cell viability assays, the AOH becomes cytotoxic above 31.3 μg/ml (Figure 3). But, in their array data summary (Table 1) they suggest they ran the arrays at 100 and 300 μg/ml which would be 3-100 fold excess of the cytotoxic levels noted. And, throughout the paper the authors appear to keep switching between μg/ml and mg/ml. These units are not interchangeable! The authors either have made significant typographical errors or, they are not reporting their data correctly.
Response 1: Figure 3 shows the cell viability after 48 hours of incubation with AOH added to the cells. The reason why the results of cell viability after 48 hours of culture are shown in the graph is because the cell activation effect of 31.3 μg/mL or lower was remarkable.
For the DNA microarray, the incubation time after the addition of AOH was set to 24 hours. The reason is that no cell damage was observed at 500 μg/mL or less when the incubation time was 24 hours.
As you pointed out, the units were wrong, and we apologize for that. All the parts where "micro" was mistakenly converted to "milli" have been corrected.
Point 2: Regarding the array work, it is unclear from the Method section how the authors determine their statistical significance. It would be inappropriate to run the same set of cells three times and report a standard deviation in the measurement. To determine array standard deviations, the authors would need to run three or more separate arrays, using the same cells, treatments and data points, and then average and determine significance. This does not appear to be what they are doing (or it is unclear that this is what they are doing). Outside of this kind of analysis on arrays, it can only be determine subjectively that the treatments are having an impact on the cell genomic expressions.
Response 2: Thank you for your remarks regarding the methods and analysis of DNA microarrays. I think you are right. However, our research budget did not allow us to run more than three separate arrays. Therefore, in this study, we performed DNA microarrays with different concentrations of AOH to understand the potential of AOH on the skin. For the genes that were found to be variable, we will need to perform RT-PCR in the future to refine the data. However, the results of the clinical trial (double-blind test) showed an increase in stratum corneum water content and a decrease in TEWL after 8 weeks of AOH application. We believe that the data from the clinical trial supports the results of the DNA microarray.

Reviewer 2 Report
You used 30, 100, and 300 µg/mL for the microarray (Table 1). However, in Figure 3, concentrations ≥ 250 µg/mL showed mild cytotoxicity. Please explain why the concentrations were selected and any limitations to assess the microarray results.
Regarding Table 1, although the upregulation was statistically significant, the magnitude of the increase was minimal. Add discussions around the interpretation. You may want to have the positive control to compare if the magnitude is still clinically meaningful. A heatmap may be more desirable.
Author Response
Thank you very much for reviewing our article. We apologize for the delay in response.
Point 1: You used 30, 100, and 300 µg/mL for the microarray (Table 1). However, in Figure 3, concentrations ≥ 250 µg/mL showed mild cytotoxicity. Please explain why the concentrations were selected and any limitations to assess the microarray results.
Response 1: Figure 3 shows the cell viability after 48 hours of incubation with AOH added to the cells. The reason why the results of cell viability after 48 hours of culture are shown in the graph is because the cell activation effect of 31.3 μg/mL or lower was remarkable.
For the DNA microarray, the incubation time after the addition of AOH was set to 24 hours. The reason is that no cell damage was observed at 500 μg/mL or less when the incubation time was 24 hours.
All the parts where "micro" was mistakenly converted to "milli" in the text have been corrected.
Point 2: Regarding Table 1, although the upregulation was statistically significant, the magnitude of the increase was minimal. Add discussions around the interpretation. You may want to have the positive control to compare if the magnitude is still clinically meaningful. A heatmap may be more desirable.
Response 2: As you pointed out, I decided that I needed positive control. However, we did not think that there was an appropriate compound to compare with AOH. We also decided that it was most important to understand the potential of AOH on the skin, so we decided to perform DNA microarrays with different concentrations of AOH in this study.
We could not show the results in a heat map because we do not own the analysis software to create heat maps.
We have already conducted a clinical trial. The results of the clinical trial showed an increase in stratum corneum water content and a decrease in TEWL after 8 weeks of AOH application. We are confident that the data from the clinical trial supports the results of the DNA microarray. The results of the clinical trial are currently under preparation for publication.
Please see the attached file.

Round 2
Reviewer 1 Report
The authors appear to have addressed my primary concerns around the mislabeled units and disconnect in the testing treatment levels verses the cytoxicity levels. I can also appreciate that running microarrays is expensive and that it is not always possible to run three arrays to gain statistical significance. I find the author's comments on this aspect of the studies to be suitable to explain their scientific reasoning.
The paper is suitable for publication in the corrected and updated form.
Reviewer 2 Report
I confirmed that the authors addressed my concern in the revised manuscript.